# Triggering Chinese lecturers' intrinsic work motivation by value-based leadership and growth mindset: Generation difference by using multigroup analysis

Xiangge Zhao[1]*, Walton Wider[2]*, Xinxin Zhang[1], Muhammad Ashraf Fauzi[3], Chee Hoo Wong[2], Leilei Jiang[4], Lester Naces Udang[5,6]

1 School of Foreign Languages for International Business, Hebei Finance University, Baoding, Hebei, China, 2 Faculty of Business and Communications, INTI International University, Nilai, Negeri Sembilan, Malaysia, 3 Faculty of Industrial Management, Universiti Malaysia Pahang Al-Sultan Abdullah, Gambang, Malaysia, 4 Faculty of Education and Liberal Arts, INTI International University, Nilai, Negeri Sembilan, Malaysia, 5 School of Liberal Arts, Metharath University, Phatum Thani, Thailand, 6 Educational Psychology, College of Education, University of the Philippines, Diliman, Philippines

* 1853657774@qq.com (XZ); walton.wider@newinti.edu.my (WW)

**Data Availability Statement:** All dataset files are available from the Zenodo database (https://doi.org/10.5281/zenodo.8404991).

## Abstract

This cross-sectional study investigated the effects of value-based leadership and growth mindset on the intrinsic work motivation of Chinese lecturers. In addition, this study used age as a categorical moderator to investigate generational differences between the effects of Millennials and their predecessors. A sample of 518 lecturers from various Chinese universities was used to collect data, and SEM-PLS was used to analyse the data. The results showed that value-based leadership and growth mindset had a significant positive impact on both younger and older lecturers' intrinsic work motivation, with the effect of value-based leadership on younger lecturers' intrinsic motivation being significantly stronger than on older lecturers' intrinsic motivation, whereas the effect of growth mindset on intrinsic work motivation did not differ significantly between the younger and older groups. This study contributes to the existing research literature by contrasting the value-based leadership and growth mindset in relation to lecturers' intrinsic work motivation across younger and older groups in Chinese higher education settings, where greater heterogeneity between age groups was identified. The findings also provided university administrators with recommendations for boosting the intrinsic work motivation of lecturers, influencing future education policy.

## Introduction

In higher education, lecturers' work motivation is critical to their engagement in teaching, academic research, and the university's long-term development [1,2]. The term "lecturer" used in this study refers to individuals who hold full time teaching positions at universities [3]. To clarify, in this context, we include both non tenure-track and tenure-track lecturers. Lecturers'

**Funding:** The author(s) received no specific funding for this work.

**Competing interests:** The authors have declared that no competing interests exist.

primary responsibilities in China's universities usually include teaching (i.e., delivering lectures, conducting seminars, and leading discussions to impart knowledge and skills to students in their respective subject areas), research (i.e., conducting research projects, publishing academic papers, and contributing to the advancement of knowledge in their field of expertise), academic advising (i.e., provide guidance and support to students, helping them with their academic and career development), curriculum development (designing and updating curriculum materials, ensuring that they align with the latest academic standards and industry requirements), and service to the institution (i.e., various administrative duties within the university, such as serving on committees, participating in faculty meetings, and contributing to the overall academic governance of the institution).

Motivated lecturers would undoubtedly result in improved student performance, higher educational quality, and even help the country grow [4]. Intrinsic motivation, which is supported by ingrained values like personal satisfaction, interest, enjoyment, or challenge, is thought to have a longer-term effect on engagement, performance, and psychological well-being than extrinsic motivation, which is fueled by outside, financial or immaterial factors like cash, promotion, or verbal praise [5,6]. However, strategies for increasing intrinsic motivation are context and culturally specific [1,7]. In contrast to Western developed countries, China is a fast rising developing country with a long history of Confucianism and values that emphasise group orientation, interpersonal harmony, submission of authority, and benevolence, but no explicit autonomy [8]. Key motivators for lecturers may differ significantly between China and Western countries [9]. It is thus critical to investigate the factors that influence lecturers' intrinsic work motivation in Chinese cultural contexts.

Furthermore, people's motives are quite complex, and they are likely to change in tandem with changes in social cultures, economic conditions, working environments, working attitudes, and values [10]. As a result, previous research has found some differences in work motivation across generations or age groups [4,11–13]. Universities should therefore consider generational differences when motivating lecturers to work. Since 1978, China's economy, culture, and society have been rapidly changing, resulting in significant shifts in personal and social values; thus, some generational differences in motivational factors are to be expected [14]. In China, those who were born and raised following 1978 economic reforms and the 'one-child policy' (now stated to be phased out as of 2016) in the 1980s and 1990s are referred to Millennials or new generation in China (see [15]). They have a stronger demand for participation as well as a different set of perspectives and preferences than prior generations of employees due to the influence of shifting economic, political, and social contexts [16]. Such demographic changes in the workplace have called into question long-held management practises [17]. However, few studies have looked at the impact of influencing factors on Chinese lecturers' intrinsic work motivation across generations, with a particular lack of focus on Millennial lecturers, who now form a substantial portion of the university workforce.

This study seeks to fill these gaps by investigating the impact of motivational factors (such as value-based leadership and a growth mindset) on Chinese lecturers' intrinsic work motivation, as well as the differences across age groups. The partial least squares-structural equation modelling (PLS-SEM) method was applied to evaluate the measurement model and structural model for different generations of lecturers, and the results were compared using multigroup analysis (MGA). By providing a comprehensive examination of the motivational factors and generational differences affecting Chinese lecturers' intrinsic work motivation, this study contributes to enriching existing knowledge in the field. The findings will not only enhance our understanding of how to effectively motivate lecturers but also inform management strategies in response to the changing workforce demographics in Chinese universities.

## Underpinning theory

Numerous theoretical developments that fall into three categories have emerged as a consequence of the research on intrinsic motivation [18]. The first category, characterized as need-motive-value, emphasises the importance of individual needs, motives, and values in generating specific behaviours [1], using guidelines from Maslow's need-hierarchy theory [19], Alderfer's [20]'s existence-relatedness-growth (ERG) theory, [21]'s equity-theory, [22]'s three-needs-theory of achievement, and [23]'s two-factor theory. These theories share similarities in their focus on understanding what motivates individuals to engage in specific behaviors, and they recognize that individuals have varying needs, motives, and values that influence their actions. Maslow's need-hierarchy theory suggests that individuals are motivated by a hierarchy of needs, starting from basic physiological needs and progressing to higher-level needs such as self-actualization. Alderfer's ERG theory proposes three core needs: existence, relatedness, and growth, asserting that individuals can be motivated by multiple needs simultaneously and that frustration of one need may lead to the emergence of another need. Adams' equity theory highlights the importance of perceived fairness in motivating behavior, suggesting that individuals compare their inputs and outcomes with those of others and strive for fairness in these exchanges. McClelland's three-needs theory focuses on the three innate needs of achievement, affiliation, and power, stating that individuals vary in their dominant need, which influences their motivation and behavior. Herzberg's two-factor theory distinguishes between hygiene factors and motivators. Hygiene factors, such as work conditions and salary, are necessary to prevent job dissatisfaction, while motivators, such as recognition and growth opportunities, lead to job satisfaction and intrinsic motivation.

The second category, emphasising cognitive choice theories, focuses on how people recognise and sustain their values, develop intrinsic motivation, and make wise judgements and logical choices, as demonstrated by [24]'s expectancy-value theory and [25]'s attribution theory. Vroom's expectancy-value theory suggests that people's motivation to engage in a certain behavior is determined by two factors: expectancy and value. Expectancy refers to an individual's belief that their efforts will lead to successful performance, while value represents the subjective importance or desirability of the outcomes associated with the behavior. According to this theory, individuals are more likely to be intrinsically motivated if they believe that their efforts will result in a successful outcome, and if they perceive the outcomes to be valuable and rewarding. Weiner's attribution theory focuses on how individuals attribute causality to their successes or failures. According to this theory, people's intrinsic motivation can be influenced by how they explain the reasons behind their performance outcomes. Weiner's theory proposes three dimensions of attributions: locus of control (internal vs. external), stability (stable vs. unstable), and controllability (controllable vs. uncontrollable). For example, if individuals attribute their success to internal factors (e.g., effort, ability) rather than external factors (e.g., luck, task difficulty), they are more likely to maintain and enhance their intrinsic motivation.

The third category, emphasising self-regulation or metacognition theories, highlights the significance of attentive goals and plans on people's intrinsic motivation, based on [26]'s control theory, [27]'s cognitive evaluation theory, [28]'s goal-setting theory, and [29]'s social learning theory. The Self-Determination Theory [SDT], developed by Edward L. Deci and Richard Ryan in 1985 [30], is central to our investigation. This comprehensive framework elucidates the nature and dynamics of human motivation, with a particular emphasis on the circumstances that foster the inherent human tendencies for proactive engagement and personal growth. Three fundamental psychological needs are prioritized within the SDT framework: competence, autonomy, and relatedness [27]. Competence addresses an individual's desire to effectively influence desired outcomes and achieve task mastery. Autonomy refers to an

individual's intrinsic desire for volitional action and self-regulation that is unfettered by external constraints [31]. Finally, relatedness emphasizes the desire for meaningful connections, emphasizing the significance of mutual understanding and care. SDT recognizes the universality of these needs, though their prominence and expression may vary depending on individual experiences, cultural contexts, and temporal factors [32]. When these needs are met adequately, intrinsic motivation increases. On the contrary, if these needs are not met, it can lead to decreased intrinsic motivation, leaving people feeling externally motivated or even unmotivated.

Motivation is further classified within SDT's paradigm as amotivation, extrinsic motivation, and intrinsic motivation [27]. Extrinsic motivation is further classified as controlled (actions influenced by external pressures) and autonomous (actions influenced by personal values and autonomy) motivations [30]. Internalization is a distinct feature of SDT, denoting the incorporation of external regulations into self-regulation, culminating in intrinsic motivation when behaviors are entirely aligned with personal interests and values [32].

SDT's theoretical foundation serves as a pivotal lens for our research. Given the emphasis on shared values and ethical grounding in value-based leadership, it is hypothesized that such a leadership approach can enhance the internalization processes inherent in SDT, with significant implications for intrinsic work motivation, particularly among distinct generational cohorts.

## Literature review

### Intrinsic work motivation of Chinese lecturers

Intrinsic motivation is frequently defined as the desire to accomplish a job or activity because one is interested in or enjoys it rather than because of any external factor [33]. Lecturers' intrinsic work motivation can be defined as a multitude of factors that motivate lecturers to choose careers as educators, keep them there, and succeed at their jobs [1]. Lecturers in China were found to be unmotivated at work and unsatisfactorily devoted to their teaching jobs [34–36]. Universities and colleges in China rely more on financial and institutional incentives to motivate lecturers, but the motivating effect is not significant and/or sustainable [37,38], partly due to the Chinese cultural context rooted in Confucianism, which emphasises human virtues, and faculty are thus expected to commit to education with a spirit of utter devotion, rather than for money [39,40].

China has a rich culture with a five-thousand-year history that is distinct from western and other eastern cultures [41]. Confucianism, as the dominant thought, is at the heart of Chinese traditional culture, shaping Chinese norms and beliefs, including work-related values [42]. In Chinese culture, Confucianism lays the groundwork for collectivism by encouraging individuals to prioritise group benefits when there is a conflict between group and individual benefits [43]. As a result, lecturers are expected to work hard to achieve the university's goals, even if this means foregoing personal benefits [38]. Furthermore, Confucianism places a premium on human virtues, and people are encouraged to act like gentlemen at all times [44]. One of the most important qualities a gentleman should possess is a disregard for wealth, which may explain why Chinese people always value morality over material gains [45]. Educators, in particular, who are regarded as having moral integrity, are expected to devote themselves to education with zeal rather than for financial gain [39,40]. As a result, in China, educators are always associated with high moral character and a fairly low requirement for wealth [38]. [46] discovered in their case study of 20 Chinese teachers that the most significant aspect impacting teachers' intrinsic motivation is the meaning they create in their teaching practice. It should not only satisfy their personal needs but also adhere to external standards. However, some

recent studies on the intersection of Confucianism and modern values, particularly in the context of the rapidly developing Chinese economies, revealed some changes of Chinese millennials values. They found Chinese millennials shared values with their Western counterparts, such as individualism and self-realization, and exhibited a more assertive Western-like style [47]. In contrast to past generations, who were more collective and more inclined to adhere to Confucianism, they were also more independent and less likely to do so [48]. Therefore, more research on effective solutions to boost lecturers' work intrinsic motivation in Chinese sociocultural contexts, particularly focusing on millennials' work motivation, is still highly anticipated [9,36,49].

## Influencing factors of intrinsic work motivation

Prior empirical studies have explored the associations between various variables and intrinsic work motivation. Leadership has been identified as a significant variable linked to intrinsic work motivation [50–52]. However, leadership styles have been observed to vary across cultures [53], and national culture can play a role in determining leadership attributes and perceived effectiveness [54]. Within China, where Confucian ideas about education exert a major influence [53], leadership values hold notable prominence, suggesting that value-based leadership might resonate well within the Chinese cultural milieu. [55] described value-based leadership as interactions founded on organizational values and ethical standards, emphasizing congruence between organizational, leader, and follower values [56]. Such alignment of values can be associated with motivating lecturers, as an alignment between individual and organizational values can lead lecturers to perceive their work as socially valuable, potentially enhancing work satisfaction [57]. As proposed by [58], Confucian-based Chinese cultural traditions could potentially complement the development of value-based leadership in educational spheres. Historically, China has placed emphasis on moral preparation for leaders, with expectations centering on qualities such as service, honesty, fairness, truth-seeking, and receptivity to criticism [59]. Nonetheless, there is limited empirical research within China examining the association between value-based leadership and lecturers' intrinsic work motivation [60].

Another variable that has garnered attention in recent years is the growth mindset [61]. Mindset reflects an individual's predominant attitudes and typical problem-solving approaches [62]. The growth mindset, in particular, represents the belief in the potential for development and enhancement of human traits, including intellect, through effort and the right strategies [63]. An individual with a growth mindset tends to exhibit a disposition towards improvement and development, aligning closely with the principles of intrinsic motivation [61,62]. Such a mindset can be particularly resonant in educational environments valuing continuous learning. Yet, limited research has probed the association of growth mindset with intrinsic work motivation among Chinese lecturers. Moreover, scarce studies have endeavored to explore the joint associations of value-based leadership and growth mindset with intrinsic work motivation for lecturers.

## Generation differences in intrinsic work motivation

Past research has indicated that age might act as a moderator, suggesting potential variations in the association between influential factors and work motivation across different age groups or generations [64,65]. Additionally, generational differences seem to be more pronounced predictors of work values than age alone [14]. There has been a growing body of research in recent years exploring the role of generational differences as moderators in various cognitive and behavioral associations within organizational and institutional contexts [66]. Furthermore, it is essential to consider recent literature on cognition and the development of cultural

norms and attributes, as it offers valuable insights into the subsequent behavioral outcomes that may arise [67]. While some Western studies have explored generational distinctions in work values [68,69], the distinct historical, social, and cultural experiences of China imply generational classifications distinct from Western paradigms. This results in differing definitions and operationalizations of generational cohorts, leading to diverse findings across studies [70]. Western research often identifies four primary generations: Traditionalists or the Silent generation (1925–1945), Baby Boomers (1946–1964), Generation X (1965–1981), and Millennials or Generation Y (1982–1999) [71]. In contrast, Chinese classifications discern three generations: the Cultural Revolution generation (born between 1949 and 1966), the Transitional generation (born between 1967 and 1978), and the Millennial generation (born between 1979 and 1990) [14,72].

Among consistent observations, younger faculty members seem to align more closely with value-based leadership emphasizing teamwork, communication, and innovation, whereas older faculty appear to resonate with traditional leadership anchored in hierarchy and convention [73–75]. Millennials, relative to older generations, seem to be more associated with environments fostering both personal and professional development [76,77] and exhibit a pronounced desire for recognition and respect [78]. The Socio-emotional Selectivity Theory [79], suggests that younger individuals, having a broader future outlook, might prioritize goals conducive to growth, while older individuals may lean towards goals enhancing immediate emotional satisfaction. However, prior research presents varied outcomes regarding age-associated differences in the relationship of growth mindset with intrinsic work motivation. Some studies suggest a more notable association among younger employees [75,80], while others find no discernible difference [81,82]. Furthermore, limited research has ventured into this domain within the context of Chinese academic institutions.

In summary, considering the aging workforce and the evolving generational landscape, understanding potential generational variations in response to distinct leadership styles and mindsets becomes imperative. This is especially relevant for China, experiencing rapid expansion in its higher education sector and notable generational transitions among faculty. Yet, the associations between value-based leadership, growth mindset, and intrinsic work motivation across varying generational groups within Chinese universities remain underexplored. Engaging in multigroup analyses could offer insights into potential generational variations in these relationships, facilitating more generation-specific interventions in academic settings.

Based on the preceding discussions, two hypotheses are proposed in this study:

H1: There is a significant difference between Millennials and their predecessors in the effect of value-based leadership on intrinsic work motivation.

H2: There is a significant difference between Millennials and their predecessors in the effect of growth mindset on intrinsic work motivation.

## Methodology

### Sample and population

In accordance with the information released by the Ministry of Education of the People's Republic of China in 2021, there were 2,740 regular Higher Education Institutions (HEIs) in China, employing a total of over 1.83 million full-time lecturers. As a result, the population of this study is 1.83 million Chinese lecturers. Using [83]'s sample size calculator, with a population of 1.83 million, a confidence level of 95%, and a confidence interval of 5, the minimum

**Table 1. Profile of the respondents.**

| | | Frequency | | Percentage (%) | |
|---|---|---|---|---|---|
| | | **Younger** | **Older** | **Younger** | **Older** |
| Gender | Male | 42 | 82 | 15.1 | 34.2 |
| | Female | 236 | 158 | 84.9 | 65.8 |
| University Types | 985/211 universities | 51 | 20 | 9.8 | 3.9 |
| | None-985/None-211 universities | 172 | 194 | 33.2 | 53.0 |
| | Private universities | 55 | 26 | 10.6 | 5.0 |
| Discipline | Formal science (mathematics, statistics, logic, etc.) | 28 | 7 | 10.1 | 2.9 |
| | Natural science (physics, chemistry, geology, biology, etc.) | 80 | 52 | 28.8 | 21.7 |
| | Applied science (engineering, medicine, etc.) | 56 | 54 | 20.1 | 22.5 |
| | Social science (economics, business, law, education, history, linguistics, etc.) | 114 | 127 | 41.0 | 52.9 |

sample size for this study should be 384. As a result, the sample size of 518 completed questionnaires answered by 278 younger lecturers (40 years old) and 240 older lecturers (> = 40 years old) was more than sufficient for the analysis. Because this study aims to investigate the generation gap between Millennials (younger group) and their predecessors (older group), who are considered generations with different work values influenced by changing economic, political, and social environments in China, the age group was divided at 40 years old [17].

Table 1 depicts the demographic profile of respondents, divided into younger and older respondents. According to the findings, younger respondents were 55.2% male and 44.8% female, whereas older respondents were 51% male and 49% female. Participants' academic affiliations ranged across a variety of university types. Universities in China are composed of public and private ones. Among them, public universities can be further divided into "985 universities", "211 universities", and "non-985/non-211" universities. "985/211 universities" are considered top universities in China. The younger group from 985/211 universities made up 9.8% of the total, while the older group from the same category made up 3.9%. There are 39 "985 universities" and 112 "211 universities" in total. The younger group made up 33.2% of the None-985/None-211 universities, while the older group made up 53.0%. Finally, 10.6% of younger respondents and 5.0% of older respondents attended private universities. The respondents were divided into four academic disciplines: formal science (mathematics, statistics, logic, etc.), 10.1% in the younger group and 2.9% in the older group; natural science (physics, chemistry, geology, biology, etc.), 28.8% in the younger group and 21.7% in the older group; applied science (engineering, medicine, etc.), 20.1% in the younger group and 22.5% in the older group; and social science (economics, business, law, education, history, linguistics, etc.), 41.0% in younger group and 52.9% in older group.

## Procedure

We employed a cross-sectional design in this study. The questionnaire survey was conducted in February 2021 using a convenience sampling technique on wjx.cn, China's most popular free online survey platform. Initially, we informed the respondents about the research objectives, their rights as participants, the potential risks and benefits of participation, and the measures taken to protect their privacy and confidentiality. The requirement for ethical approval was waived by the ethics committee due to the low-risk nature of the study and the anonymized method of data collection, ensuring that no personal identifiers were collected or stored. Before proceeding with the questionnaire, they were required to give their written consent. The survey received 565 responses, and 518 cases were retained for data analysis after

removing straightliners during data cleaning, because straight lining responses were thought to reduce variability and cause undetected (or detected but underestimated) moderating effects in multigroup analysis [84]. Furthermore, comparable sample sizes for each group were recommended in order to maximise sample variance, since imbalanced sample sizes across moderator-based subgroups would limit statistical power and cause underestimation of moderating effects [85].

## Measures

Intrinsic work motivation was measured using a 3-item scale modified from the Chinese version of the Multidimensional Work Motivation Scale (MWMS) by [86]. To assess value-based leadership, an updated 19-item scale by [87] was used. A three-item scale by [63] was used to assess the growth mindset of lecturers. All of the aforementioned measures in this study scored responses on a 7-point Likert scale that ranged from 1 (strongly disagree) to 7 (strongly agree).

Translation was needed because the targeted population of this study was lecturers in Chinese universities. In order to enhance the translation validity, the translated questionnaire was sent to three Chinese professors in the domain of English Linguistics, who were asked to do a translation test and fill in a comment form by responding to the statements: *(a) Do you think the Chinese version accurately conveys the original meaning of the English version? (b) Is the phrasing and terminology in Chinese clear and easy to understand? (c) Is there any important background information that may be missing? (d) Please include any other comments relevant to the improvement of survey translation.* Overall, they gave consistent response in the translation test, confirming the Chinese version accurately conveyed the original meaning of the English version and the phrasing and terminology in Chinese are clear and easy to understand without missing any important background information.

## Data analysis

SmartPLS 3.2.9 was used to evaluate both measurement and structural models, as well as to conduct a multi-group analysis (MGA) to compare the effects of value-based leadership and growth mindset on intrinsic work motivation in younger and older age groups. PLS-SEM was used because non-parametric SEM is better suited for MGA [85]. This study evaluated the measurement model by assessing the reliability and validity of reflective constructs, as well as the structural model by evaluating the $R^2$, $Q^2$, and path coefficients [88]. Following the assessment of the measurement and structural model, MGA was performed using two different non-parametric methods, Henseler's MGA [89], and the permutation test [90]. Furthermore, measurement invariance was assessed prior to performing the MGA using the measurement invariance for composite (MICOM) approach [89].

The Common Method Variance (CMV) was checked in this study using the full collinearity variance inflation factor (VIF) [91]. When using the PLS-SEM to check the CMV, the literature recommends using full collinearity VIF and a threshold of 5 [92]. The full collinearity VIF of all constructs in the current study was 1.354, indicating a model free of CMV.

## Results

### Measurement model assessment

We evaluated the measurement and structural models for both younger and older groups of lecturers using PLS-SEM. This study's conceptual framework comprised three reflective constructs: intrinsic work motivation (IWM), value-based leadership (VBL), and growth mindset (GM). To evaluate the measurement model, the indicator and construct reliability, convergent

validity, and discriminant validity of these three reflective constructs for lecturers of both younger and older ages were evaluated [85]. To establish indicator reliability, the outer loading of the indicators for each construct must exceed 0.70. To demonstrate construct reliability and convergent validity, the Composite Reliability (CR) and Cronbach's alpha must be greater than 0.7, and the Average Variance Extracted (AVE) must be higher than 0.5. [85]. Table 2 demonstrates that the reliability and convergent validity of all three constructs in this study were acceptable for lecturers of all ages. To prove discriminant validity, we used the most conservative approach, the hetero-trait-monotrait (HTMT) ratio, which must be less than 0.90 [93]. The most recent research suggests that studies should rely solely on the HTMT criterion and use bootstrapping to determine whether its values significantly deviate from a predetermined threshold [94]. The results of HTMT 0.90 are presented in Table 3, indicating that discriminant validity is acceptable for both younger and older groups of lecturers.

**Table 2. Results of measurement model assessment.**

| Construct/Item | Cronbach's alpha | | CR | | AVE | | Loading | |
|---|---|---|---|---|---|---|---|---|
| | Younger | Older | Younger | Older | Younger | Older | Younger | Older |
| **Intrinsic Work Motivation** | 0.888 | 0.916 | 0.931 | 0.947 | 0.817 | 0.856 | | |
| IWM1 | | | | | | | 0.914 | 0.920 |
| IWM2 | | | | | | | 0.912 | 0.949 |
| IWM3 | | | | | | | 0.908 | 0.906 |
| **Value-based Leadership** | 0.975 | 0.982 | 0.977 | 0.984 | 0.687 | 0.763 | | |
| VBL1 | | | | | | | 0.846 | 0.879 |
| VBL2 | | | | | | | 0.863 | 0.891 |
| VBL3 | | | | | | | 0.823 | 0.843 |
| VBL4 | | | | | | | 0.822 | 0.840 |
| VBL5 | | | | | | | 0.817 | 0.800 |
| VBL6 | | | | | | | 0.877 | 0.901 |
| VBL7 | | | | | | | 0.844 | 0.871 |
| VBL8 | | | | | | | 0.882 | 0.895 |
| VBL9 | | | | | | | 0.888 | 0.901 |
| VBL10 | | | | | | | 0.852 | 0.866 |
| VBL11 | | | | | | | 0.891 | 0.905 |
| VBL12 | | | | | | | 0.873 | 0.916 |
| VBL13 | | | | | | | 0.894 | 0.922 |
| VBL14 | | | | | | | 0.880 | 0.912 |
| VBL15 | | | | | | | 0.711 | 0.699 |
| VBL16 | | | | | | | 0.824 | 0.847 |
| VBL17 | | | | | | | 0.859 | 0.891 |
| VBL18 | | | | | | | 0.863 | 0.904 |
| VBL19 | | | | | | | 0.850 | 0.883 |
| **Growth Mindset** | 0.824 | 0.855 | 0.895 | 0.911 | 0.739 | 0.774 | | |
| GM1 | | | | | | | 0.842 | 0.844 |
| GM2 | | | | | | | 0.874 | 0.903 |
| GM3 | | | | | | | 0.889 | 0.891 |

Note: See Appendix for complete item names.

**Table 3. Discriminant validity (HTMT$_{0.9}$ criterion).**

| | Younger | | | Older | | |
|---|---|---|---|---|---|---|
| | **GM** | **IWM** | **VBL** | **GM** | **IWM** | **VBL** |
| GM | | | | | | |
| IWM | 0.570 | | | 0.476 | | |
| VBL | 0.655 | 0.685 | | 0.473 | 0.437 | |

## Structural model assessment

To evaluate the structural model, the $R^2$ and Stone-Geisser criterion ($Q^2$) for intrinsic work motivation, as well as the significance of the path coefficients for the two groups, must be calculated [85]. The results revealed $R^2$ values of 0.43 for younger lecturers' intrinsic work motivation and 0.26 for older lecturers' intrinsic work motivation. Compared to the criterion proposed by [90], predictive power values of approximately 0.67, 0.33, and 0.19 are regarded as substantial, moderate, and weak, respectively. Therefore, the $R^2$ values indicated the structural model's moderate quality. To demonstrate the predictive capability of a structural model, the value of $Q^2$ must be greater than zero [85]. In this instance, we observed Q2 values of 0.344 and 0.201 for younger and older groups, respectively, indicating the model's predictive validity. In addition, it is essential to determine the significance of the path coefficient using bias-corrected (BCa) confidence intervals [88]. Table 5 demonstrates the positive and statistically significant effects of value-based leadership and growth mindset on the intrinsic motivation of younger and older lecturers. The path coefficient of value-based leadership on intrinsic work motivation was 0.541 for younger lecturers and 0.291 for older lecturers. Similarly, the path coefficients of growth mindset on intrinsic work motivation were 0.172 and 0.307 for younger and older lecturers respectively.

## Multigroup analysis (MGA)

Recent methodological research has paid considerable attention to the detection and handling of heterogeneity [95,96], but few studies focusing on categorical moderators have used the MICOM procedure to address the measurement invariance issue [94]. This practise was deemed troublesome because establishing measurement invariance is a prerequisite for multigroup analysis and ensures that group differences in model estimates are not caused by group-specific response patterns [97]. MICOM procedure [89], consists of three steps: (a) configural invariance, (b) compositional invariance, and (c) the equality of the mean value and variance of a composite across groups. In this study, configural invariance is established with the same indicators in both younger and older groups when evaluating the reliability and validity of the data, data preparation, and SmartPLS algorithm settings (e.g., path weighting with a maximum of 300 iterations and a stop criterion of $10^{-7}$). Concerning the evaluation of compositional invariance, the results of correlation between the composite scores of the younger and older groups were compared with the 5% quantile, which revealed that the quantile was less than (or equal to) correlation for all constructs, indicating that compositional invariance was established. The establishment of partial measurement invariance was indicated by the fulfilment of both the configural and compositional invariance assessment criteria. In light of the MICOM results in Table 4, MGA can be used to compare the path coefficients for both groups and test the hypotheses. Table 5 displays the results of MGA using both nonparametric methods: Henseler's MGA [89], and the permutation test [90], which are considered the most conservative PLS-SEM techniques for assessing differences in path coefficients between two groups [94]. In

**Table 4. Results of invariance measurement testing using permutation.**

| Constructs | Configural invariance | Compositional invariance (Correlation = 1) | | Partial measurement invariance established | Equal mean assessment | | | Equal variance assessment | | | Full measurement invariance established |
|---|---|---|---|---|---|---|---|---|---|---|---|
| | | C = 1 | Confidence interval | | Differences (Younger-Older) | Confidence interval | Equal | Differences (Younger-Older) | Confidence interval | Equal | |
| GM | Yes | 0.999 | [0.996, 1.000] | Yes | 0.230 | [-0.162, 0.154] | No | -0.077 | [-0.256, 0.247] | Yes | No |
| IWM | Yes | 1.000 | [0.999, 1.000] | Yes | -0.162 | [-0.153, 0.151] | No | 0.092 | [-0.277, 0.228] | Yes | No |
| VBL | Yes | 1.000 | [0.999, 1.000] | Yes | 0.102 | [-0.166, 0.166] | Yes | -0.327 | [-0.284, 0.269] | No | No |

hypothesis 1, the path coefficient shows that both are significant, with the younger group (0.541) having a higher value compared to the old group (0.291). This resulted in a difference of 0.251 path coefficient. This empirical evidence suggests that the younger group have a substantial positive relationship between value-based leadership and intrinsic work motivation. On the other hand, hypothesis 2 suggests that the path coefficient of growth mindset on intrinsic motivation is higher in the old group (0.307) compared to the younger group (0.172), resulting in 0.135 differences. This indicates that the younger group are more affected by value-based leadership, whereas the older group are more affected by a growth mindset towards intrinsic motivation. Age as a moderating variable has shown a considerable impact on variables that can consequently produce a generation gap, such as leadership and mindset, as shown by this study. The younger generation needs leaders who can guide them and act as role models in academia. In contrast, the older generation is well-equipped with a strong mindset that can spearhead changes and advancement.

## Discussion

PLS-SEM and MGA results supported H1, suggesting an association between Millennials and their predecessors in terms of how value-based leadership relates to intrinsic work motivation. Specifically, the findings indicate a stronger association of value-based leadership with intrinsic work motivation for younger lecturers than for older lecturers. This is consistent with previous research discussing the values and preferences of Millennials, who prioritize finding meaning and purpose in their work and tend to align more closely with leaders reflecting these values [48]. Raised in an era of ubiquitous information and technology, Millennials tend to

**Table 5. Results of hypothesis testing.**

| Hypothesis | Relationships | Path coefficient | | Confidence interval (95%) | | Path coefficient difference | *p*-value difference | | Supported |
|---|---|---|---|---|---|---|---|---|---|
| | | Younger | Older | Younger | Older | | Henseler's MGA | Permutation | |
| Hypothesis 1 | VBL→ IWM | 0.541*** | 0.291*** | [0.413, 0.647] | [0.153,0.431] | 0.251 | 0.004** | 0.001* | Yes |
| Hypothesis 2 | GM → IWM | 0.172** | 0.307*** | [0.056, 0.288] | [0.181, 0.426] | -0.135 | 0.938 | -0.002 | No |

Note: In Henseler's MGA technique, the *p* value below 0.05 or above 0.95 indicates at the 5% level significant differences between particular path coefficients across two groups.

*$p<0.05$,

**$p<0.01$,

***$P<0.001$.

gravitate towards work environments congruent with their values and beliefs and have shown a lower tolerance for leaders not resonating with their values [16,78]. They often exhibit confidence, feeling unthreatened by superiors, and find satisfaction in sharing their ideas with management, fostering a sense of belonging [12]. Hence, leaders and workplaces valuing their perspectives might have a stronger association with motivating Millennials. Value-based leadership, described as leaders embodying and promoting organizational core values [56], seems to have a more pronounced association with intrinsic motivation among Millennials, possibly due to their emphasis on congruence between individual and organizational values.

Previous research suggested that Millennials, being better educated and more receptive to challenges than older cohorts in this study [14,17], would exhibit a stronger association with growth-oriented beliefs (H2). However, H2 was not supported as there was no notable distinction in the association of growth mindset with intrinsic work motivation between younger and older lecturers. This could be attributed to the overarching value placed on a growth mindset among educators of all ages, especially within the Chinese context that prioritizes continuous learning as a pathway for personal enhancement [58,98]. This observation underscores the importance of integrating cultural contexts when analyzing motivational theories and their relevance across different demographics. It hints that in cultures, such as China, emphasizing life-long learning, the association of age with goal orientation might be less distinctive than in regions influenced by different dynamics.

## Implications and limitations

Theoretically, this study adds to existing motivation literature by demonstrating that value-based leadership and a growth mindset have significant effects on the intrinsic work motivation of both younger and older lecturers in the Chinese socio-cultural context. Furthermore, the identification of generational differences in the effects of value-based leadership on intrinsic work motivation suggests that when examining motivational factors, more attention should be paid to the heterogeneity of different age groups. However, the explanations for generational differences in leadership are fragmented and mostly prescriptive [99], necessitating more systematic future research.

In practise, this research contributes to a better understanding of how to boost lecturers' intrinsic work motivation in higher education settings. The findings suggest that university administrators can increase the intrinsic motivation of lecturers by developing value-based leadership and lecturers' growth mindset. This necessitates the leaders' ability to effectively communicate the organisational values to members, ensuring that the organisational values are widely accepted by members and serve as the foundation for a harmonious working environment. University leaders should also make an effort to develop lecturers' growth mindsets, giving them more opportunities to expand their capabilities at work and grow through regular reflection. To maintain a consistent and meaningful routine, one solution is to incorporate reflection activities into professional development or faculty meetings [100]. Particularly for Millennial lecturers, their leaders should be more concerned with their work values, serve as a guide for them at all levels, and foster supportive relationships among them with a united, open-hearted, and shared vision [101,102], so that they can be more intrinsically motivated and achieve long-term development.

While this study provides valuable insights into the relationship between value-based leadership, growth mindset, and intrinsic work motivation among Chinese lecturers, there are several limitations that should be noted. Firstly, it is important to acknowledge that other demographic variables may also have moderating effects on the relationship between value-based leadership, growth mindset, and intrinsic work motivation among Chinese lecturers.

For example, recent studies by [103] have shown that gender can play a significant role in influencing cognitive and behavioral outcomes. Therefore, future research could explore the potential influence of gender as a moderating variable in this context. Furthermore, considering the impact of individuals' self-efficacy on managerial behaviors in higher education, as highlighted by [103], could provide valuable insights into the dynamics between value-based leadership, growth mindset, and intrinsic work motivation among lecturers. Incorporating self-efficacy as a determinant in future studies could enhance our understanding of the complex interplay of factors affecting lecturers' motivation. It is also worth noting that the convenience sampling method used due to the COVID-19 pandemic may limit the generalizability of the research findings [104]. Additionally, the study's narrow focus on Chinese universities, which are influenced by China's governmental system, bureaucratic mode, and Confucius-rooted values [105], may restrict the generalizability of the results to other nations with different cultural contexts.

## Conclusion

This study was motivated by a significant challenge faced by lecturers, specifically the rapid development of higher education in China. Due to the intense, high-pressure, and rapid work environment, many lecturers report physical and mental exhaustion, which correlates to a decline in their work motivation [106]. This study sought to empirically explore the relationships between value-based leadership (an organizational factor) and growth mindset (an individual factor) with lecturers' intrinsic work motivation, with an emphasis on potential differences across age groups. The observed significant difference in the association between value-based leadership and intrinsic work motivation across millennials and their preceding generations highlighted the potential moderating role of generational factors. Yet, this study primarily emphasized the contrast between millennials and their predecessors, even though there's extensive literature pointing to Generation Z (born post-1995) as a noteworthy emerging demographic [64,107]. In China, while Generation Z lecturers currently represent a minority, it would be intriguing for future research to investigate the motivational aspects as their representation increases. Beyond age, additional variables like gender, income, and discipline could be explored as potential moderators in upcoming studies. This might provide insights into variations in the associations of interventions with lecturers' work motivation, possibly merging them to discern intricate variations. Through such exploration, strategies to engage lecturers' work motivation might be crafted more specifically, enhancing their potential effectiveness.

## Supporting information

**S1 File.**
(DOCX)

## Author Contributions

**Conceptualization:** Xiangge Zhao, Walton Wider, Chee Hoo Wong.

**Data curation:** Xiangge Zhao.

**Formal analysis:** Xiangge Zhao, Walton Wider.

**Funding acquisition:** Xiangge Zhao.

**Investigation:** Xiangge Zhao.

**Methodology:** Xiangge Zhao, Walton Wider.

**Project administration:** Xiangge Zhao.

**Resources:** Xiangge Zhao.

**Software:** Xiangge Zhao, Walton Wider.

**Validation:** Xiangge Zhao, Walton Wider.

**Visualization:** Xiangge Zhao, Walton Wider, Chee Hoo Wong.

**Writing – original draft:** Xiangge Zhao.

**Writing – review & editing:** Walton Wider, Xinxin Zhang, Muhammad Ashraf Fauzi, Chee Hoo Wong, Leilei Jiang, Lester Naces Udang.

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
