## [Decision Letter · Decision Letter 0]

29 Aug 2023

PONE-D-23-19053Triggering Chinese lecturers’ intrinsic work motivation by value-based leadership and growth mindset: Generation difference by using multigroup analysisPLOS ONE

Dear Dr. Wider,

Thank you for submitting your manuscript to PLOS ONE. After careful consideration, we feel that it has merit but does not fully meet PLOS ONE’s publication criteria as it currently stands. Therefore, we invite you to submit a revised version of the manuscript that addresses the points raised during the review process.

ACADEMIC EDITOR: Dear authors: I send you the comments of the reviewers. I agree with the suggested changes to improve the study. It is important that you read them carefully and incorporate their recommendations into the manuscript. Please respond clearly to every point raised by the reviewers. Best regards

We look forward to receiving your revised manuscript.

Kind regards,

Alejandro Ros Gálvez

Academic Editor

PLOS ONE

Also, you indicated that ethical approval was not necessary for your study. We understand that the framework for ethical oversight requirements for studies of this type may differ depending on the setting and we would appreciate some further clarification regarding your research. Could you please provide further details on why your study is exempt from the need for approval and confirmation from your institutional review board or research ethics committee (e.g., in the form of a letter or email correspondence) that ethics review was not necessary for this study? Please include a copy of the correspondence as an ""Other"" file.

Reviewers' comments:

Reviewer's Responses to Questions

**Comments to the Author**

1. Is the manuscript technically sound, and do the data support the conclusions?

Reviewer #1: Yes

Reviewer #2: Yes

2. Has the statistical analysis been performed appropriately and rigorously? 

Reviewer #1: Yes

Reviewer #2: Yes

3. Have the authors made all data underlying the findings in their manuscript fully available?

Reviewer #1: Yes

Reviewer #2: Yes

4. Is the manuscript presented in an intelligible fashion and written in standard English?

Reviewer #1: Yes

Reviewer #2: Yes

5. Review Comments to the Author

Reviewer #1: This study attempts to address the effects of value-based leadership and growth mindset on the intrinsic work motivation of Chinese lecturers as well as investigate the generational differences between the effects of Millennials and their predecessors. The manuscript is overall well written, presents a clear theoretical framework, methodology, findings and discussions for the study. However, it left some to be desired and needs minor revision, especially on its methodology section. See comments on the attachment.

Reviewer #2: Thank you for this opportunity to read this study. The theme is topical, but several concerns have to be solved before re-evaluating the manuscript and possibly considering it for publication.

As general observation, this study is a cross-sectional study and it is incorrect to use causal term in description of the relations between variables. Please revise all the manuscript and replace all the causal terms.

1. The author should try to add the most critical literature (2019-2023) to the topic, deeply demonstrate an understanding of this topic, critically review the literature, and point out limitations and conflicts.

2. Please integrate the Self-Determination Theory (Deci & Ryan, 2008) as theoretical background to explain the relations between hypothetical model.

3. Because is a cross-sectional design, it is necessary to verified the common method variance error.

4. In Discussion section, please follow the logic of each hypothesis and discuss the results of each hypothesis in relation with theoretical perspective.

6. PLOS authors have the option to publish the peer review history of their article (what does this mean?). If published, this will include your full peer review and any attached files.

Reviewer #1: No

Reviewer #2: No

---

## [Author Response · Author response to Decision Letter 0]

4 Oct 2023

Reviewer #1:

1. Since the study is about lecturers’ motivation, the term lecturer should be defined and explained. I notice that in a later section, “faculty member” is also used. Are they similar or different? Are these lecturers non tenure-track or tenure-track? Full-time or part-time? What are their primary responsibilities in the universities? Also, are these lecturers from different universities you surveyed have the same responsibilities? Because these statuses would affect their motivation, I think it needs to be addressed clearly. 

Response: In this study, we use the term "lecturer" to refer to individuals who hold teaching positions at universities. To clarify, in this context, we include both non tenure-track and tenure-track lecturers. The majority of participants in our study (84.5%) were from public universities, where they generally hold tenure-track positions; while a smaller proportion of respondents (14.5%) were from private universities, where lecturers are full-time without tenure-track positions. 

Lecturers' primary responsibilities in Chinese universities include teaching, research, academic advising, curriculum development, and service to the institution. We have provided a clearer explanation in this respect in our revised version.

We apologize for any confusion caused by the use of the term "faculty member" in a later section. In our study, we use it interchangeably with the term "lecturer" to refer to the same group of individuals.

2. In the literature review section, I would like to see some new studies on the motivation, as the context could change over time. Looking at the first paragraph on Page 3, some studies were done in 1972, 2000, 2001. Though I do think the Confucianism does influence how people think and behave, individuals may have different interpretations of the traditional thoughts. I would like to see if there are any regional, generational, and gender differences on desiring “high moral character and a fairly low requirement for wealth”. 

Response: We acknowledge your comments, and some recent studies on the intersection of Confucianism and modern values, particularly in the context of the rapidly developing Chinese economies, have been added in the revised version. 

3. On page 3, influencing factors of intrinsic work motivation are summarized and categorized, but no citations have been provided to different theories. Please add. 

Response: Citations have been added.

4. For the section on “influencing factors of ….”, I would recommend reorganize it to focus on value-based leadership rather than “organizational variables”. The authors are trying to make a connection between organizational values and values of leaders, but such connection is not explicitly stated and explained. 

Response: We acknowledge and agree with your comments and suggestions. We focus on value-based leadership rather than “organizational variables” by reorganizing it in the revised version.

5. For the methodology section, the sample size of 518 lecturers is categorized as younger (40 years old) and older (>40 years old), but in the literature review, the millennials are defined as people who were born between 1979-1990, so I would think the old-young category should be based on the definition. Is that correct? Why do the authors select 40 as the dividing line? 

Response: First, the survey in this study was conducted in February 2021, but we made the typographical error stating that the survey was conducted in February 2022 instead of February 2021. We apologize for this inadvertent mistake and have corrected it in the revision.

Second, the survey was conducted in February 2021, and based on the time of the survey, individuals born between 1979-1990 would have been 41 years old or older. However, in order to facilitate a more convenient and simpler division, we chose to use 40 years old as the age cutoff. While this may not strictly align with the definition of millennials, it allowed us to create two distinct age groups for the analysis. We acknowledge that this decision could be seen as arbitrary, but we believed that using a whole number like 40 would contribute to the ease of interpretation and analysis of the results. We appreciate your attention to this detail and make sure to clarify this explanation in the revised manuscript.

6. I am not clear about what type of universities the authors selected to survey their lecturers. Are all of them from the same university? In China, a 985 university is quite different from a college in a small city? Without knowing this type of information, it would be difficulty to draw a valid conclusion. 

Response: Universities in China are composed of public and private ones. Among them, public universities can be further divided into “985 universities”, “211 universities”, and "non-985/non-211" universities. In this study, 26 (6.6%) participants come from 985/211 universities, 207 (77.9%) of them come from non-985/non-211" universities, and 61(15.5%) are from private universities. No significant difference was found among these three types of lecturers’ intrinsic work motivation in this study. The relevant information has been added in the revision.

[Note: “985/211 universities” are considered top universities in China. There are 39 “985 universities” and 112 “211 universities” in total.]

7. As far as the measure are concerned, I found quite interesting that the authors used MWMS and Dweck’s three-item scale to do the study. The authors kept mentioning that the Chinese context is quite different from the western context to rationalize the importance and significance of the study, why didn’t they develop their own measures to reflect the Chinese context, but rather use the measures developed by western scholars? Please provide the rationales. If the measures are updated version, please provide any information on what have been updated and why? 

Response: Thank you for your feedback. In response, we utilized the culturally-adapted Chinese version of the MWMS and undertook a rigorous translation process for Dweck’s three-item scale to ensure its relevance to the Chinese context. The choice of these validated scales ensures cultural appropriateness while allowing comparability with Western studies, offering a balance between cultural relevance and scientific rigor. We'll provide further clarity in our revised manuscript.

Reviewer #2:

1. As general observation, this study is a cross-sectional study and it is incorrect to use causal term in description of the relations between variables. Please revise all the manuscript and replace all the causal terms.

Response: We have revised the manuscript and have ensured only associations, rather than causal relationships, are implied between variables.

2. The author should try to add the most critical literature (2019-2023) to the topic, deeply demonstrate an understanding of this topic, critically review the literature, and point out limitations and conflicts. 

Response: we have focused on incorporating the critical literature published between 2019 and 2023, and critically analyzed the literature, highlighting both its strengths and limitations. This allowed us to provide a well-rounded perspective and address any conflicts or gaps identified within the existing research.

3. Please integrate the Self-Determination Theory (Deci & Ryan, 2008) as theoretical background to explain the relations between hypothetical model. 

Response: We have integrated the self-determination theory.

4. Because is a cross-sectional design, it is necessary to verified the common method variance error. 

Response: We have included the CMV in the data analysis section.

5. In Discussion section, please follow the logic of each hypothesis and discuss the results of each hypothesis in relation with theoretical perspective. 

Response: In the Discussion section, we have followed the logic of each hypothesis and thoroughly discuss the results of each hypothesis in relation to the theoretical perspective.

---

## [Decision Letter · Decision Letter 1]

27 Dec 2023

PONE-D-23-19053R1Triggering Chinese lecturers’ intrinsic work motivation by value-based leadership and growth mindset: Generation difference by using multigroup analysisPLOS ONE

Dear Dr. Wider,

Thank you for submitting your manuscript to PLOS ONE. After careful consideration, we feel that it has merit but does not fully meet PLOS ONE’s publication criteria as it currently stands. Therefore, we invite you to submit a revised version of the manuscript that addresses the points raised during the review process.

We look forward to receiving your revised manuscript.

Kind regards,

Ali B. Mahmoud, Ph.D.

Academic Editor

PLOS ONE

Journal Requirements:

Reviewers' comments:

Reviewer's Responses to Questions

**Comments to the Author**

1. If the authors have adequately addressed your comments raised in a previous round of review and you feel that this manuscript is now acceptable for publication, you may indicate that here to bypass the “Comments to the Author” section, enter your conflict of interest statement in the “Confidential to Editor” section, and submit your "Accept" recommendation.

Reviewer #1: All comments have been addressed

Reviewer #2: All comments have been addressed

Reviewer #3: (No Response)

2. Is the manuscript technically sound, and do the data support the conclusions?

Reviewer #1: Yes

Reviewer #2: Yes

Reviewer #3: Yes

3. Has the statistical analysis been performed appropriately and rigorously? 

Reviewer #1: Yes

Reviewer #2: Yes

Reviewer #3: Yes

4. Have the authors made all data underlying the findings in their manuscript fully available?

Reviewer #1: Yes

Reviewer #2: Yes

Reviewer #3: (No Response)

5. Is the manuscript presented in an intelligible fashion and written in standard English?

Reviewer #1: Yes

Reviewer #2: Yes

Reviewer #3: Yes

6. Review Comments to the Author

Reviewer #1: The authors have clearly addressed my concerns. My recommendation is to accept it for publication. In the future, I hope authors would include some voices from these lecturers and let them share their motivational aspects.

Reviewer #2: Congratulations! All the recommendations were integrated in the manuscript. The manuscript could be published in this form.

Reviewer #3: In this paper, the authors investigate the effects of value-based leadership and growth mindset on the intrinsic work motivation of Chinese lecturers, while considering age as a moderating variable. Their examination of 518 university lecturers in China indicates that value-based leadership and growth mindset has a significant positive impact on both younger and older lecturers’ intrinsic work motivation. The study also shows that the effect of value-based leadership on intrinsic motivation is significantly stronger for younger lecturers. I believe that this is an interesting and relevant paper and an important area for academic discourse. The paper also benefits from good organization and clear writing. While the current draft of the manuscript is fairly well developed, it nevertheless requires additional (relatively minor) revisions/additions, which are manageable and can be executed in a reasonable amount of time. Accordingly, I would recommend directing the authors to address the remaining outlined points.

Comment 1. Introduction section: while the authors present a well-structured introduction, I suggest expanding the discussion of relevant contributions (at the bottom of page 2), and complementing it with the summary of the key findings, while articulating how these findings expand existing knowledge in a meaningful way.

Comment 2. Underpinning Theory section: While the authors list several theories that are relevant to the subject matter of this paper, the key claims of each of these theories need to be explained in somewhat greater detail. Specifically, the “first” and “second” categories of relevant theoretical developments (on page 3) require additional explanations. This should allow the authors to better position their study relative to existing conceptual stipulations. It will also allow the authors to better explain the novelty and theoretical contribution of this study.

Comment 3. Literature Review section: While the authors present a good and relevant literature review, this discussion is missing important recent studies that should be acknowledged. Specifically, generational differences have been recently examined as moderators on various cognitive and behavioral associations in organizational and institutional contexts (Mahmoud, Berman, Reisel, Fuxman, & Hack-Polay, 2023). Furthermore, since the study examines the relevance of cultural norms and values in China, I suggest citing relevant recent literature on cognition, the development of cultural norms and attributes, and the consequent behavioral outcomes (Berman, Mudambi, & Shoham, 2022).

Comment 4. Results section: While I consider the utilized methodology to be appropriate and robust, I suggest providing “economic” interpretation of several key coefficients that are discussed in this section. In particular, the economic meaning of the differences in the path coefficients between two groups found in the Multigroup Analysis (MGA) should be explained, in order to better articulate the relevance of the generational differences as a moderator.

Comment 5. Implications and Limitations section: Additionally, while the authors consider generational differences as a moderating variable in the study, other potentially relevant demographic variables are not explicitly considered. For example, gender has been recently found to be an important moderating variable between cognitive and behavioral outcomes (Mahmoud, Berman, Grigoriou, & Solakis, 2023). Similarly, individuals’ “self-efficacy” has been recently examined as a determinant of certain managerial behaviors in higher education (Mahmoud, Berman, Reisel, Fuxman, Grigoriou, & Tehseen, 2023). I suggest acknowledging these limitations in the related section.

In summary, I believe that this is an interesting and relevant paper with the potential to make a notable contribution. I wish the authors the best of luck in the further development of the paper.

References:

Mahmoud, A. B., Berman, A., Reisel, W., Fuxman, L., & Hack‐Polay, D. (2023). Examining generational differences as a moderator of extreme‐context perception and its impact on work alienation organizational outcomes: Implications for the workplace and remote work transformation. Scandinavian Journal of Psychology.

Berman, A., Mudambi, R., & Shoham, A. (2022). Linguistic structures and innovation: A behavioral approach. Journal of International Management, 28(4), 100943.

Mahmoud, A. B., Berman, A., Grigoriou, N., & Solakis, K. (2023). Investigating gender as a moderator of extreme-context perception effects on behavioural tendencies towards fashion brands on Instagram in West Africa. Journal of Marketing Communications, 1-23.

Mahmoud, A. B., Berman, A., Reisel, W., Fuxman, L., Grigoriou, N., & Tehseen, S. Entrepreneurial Marketing Intentions and Behaviours among Students: Investigating the Roles of Entrepreneurial Skills, Entrepreneurial Self-efficacy, and Family Business Exposure.

7. PLOS authors have the option to publish the peer review history of their article (what does this mean?). If published, this will include your full peer review and any attached files.

Reviewer #1: No

Reviewer #2: No

Reviewer #3: No

---

## [Author Response · Author response to Decision Letter 1]

8 Jan 2024

1. Introduction section: while the authors present a well-structured introduction, I suggest expanding the discussion of relevant contributions (at the bottom of page 2), and complementing it with the summary of the key findings, while articulating how these findings expand existing knowledge in a meaningful way. 

Answer: Thank you for your comments. We have revised the introduction to include a more comprehensive discussion of the relevant contributions and clearly articulate how our findings expand existing knowledge in a meaningful way. This will help readers better understand the significance of our research within the field.

2. Underpinning Theory section: While the authors list several theories that are relevant to the subject matter of this paper, the key claims of each of these theories need to be explained in somewhat greater detail. Specifically, the “first” and “second” categories of relevant theoretical developments (on page 3) require additional explanations. This should allow the authors to better position their study relative to existing conceptual stipulations. It will also allow the authors to better explain the novelty and theoretical contribution of this study. 

Answer: Thank you for your comments. We have revised the Underpinning Theory section to provide a more comprehensive overview of these theories and their key claims.

3. Literature Review section: While the authors present a good and relevant literature review, this discussion is missing important recent studies that should be acknowledged. Specifically, generational differences have been recently examined as moderators on various cognitive and behavioral associations in organizational and institutional contexts (Mahmoud, Berman, Reisel, Fuxman, & Hack-Polay, 2023). Furthermore, since the study examines the relevance of cultural norms and values in China, I suggest citing relevant recent literature on cognition, the development of cultural norms and attributes, and the consequent behavioral outcomes (Berman, Mudambi, & Shoham, 2022). Answer: Thank you for your suggestions. We have revised the literature review by adding the recent studies you mentioned.

4. Results section: While I consider the utilized methodology to be appropriate and robust, I suggest providing “economic” interpretation of several key coefficients that are discussed in this section. In particular, the economic meaning of the differences in the path coefficients between two groups found in the Multigroup Analysis (MGA) should be explained, in order to better articulate the relevance of the generational differences as a moderator. 

Answer: We appreciate the suggestion. We have added a discussion on the MGA result with respect to path coefficients between the two groups.

5. Implications and Limitations section: Additionally, while the authors consider generational differences as a moderating variable in the study, other potentially relevant demographic variables are not explicitly considered. For example, gender has been recently found to be an important moderating variable between cognitive and behavioral outcomes (Mahmoud, Berman, Grigoriou, & Solakis, 2023). Similarly, individuals’ “self-efficacy” has been recently examined as a determinant of certain managerial behaviors in higher education (Mahmoud, Berman, Reisel, Fuxman, Grigoriou, & Tehseen, 2023). I suggest acknowledging these limitations in the related section. 

Answer: We appreciate your suggestions to acknowledge these limitations in the related section. To address this limitation, we have included a discussion, acknowledging the omission of these relevant variables and their potential impact on the findings.

---

## [Decision Letter · Decision Letter 2]

15 Jan 2024

Triggering Chinese lecturers’ intrinsic work motivation by value-based leadership and growth mindset: Generation difference by using multigroup analysis

PONE-D-23-19053R2

Dear Dr. Wider,

We’re pleased to inform you that your manuscript has been judged scientifically suitable for publication and will be formally accepted for publication once it meets all outstanding technical requirements.

Kind regards,

Ali B. Mahmoud, Ph.D.

Academic Editor

PLOS ONE

Additional Editor Comments (optional):

Reviewers' comments:

Reviewer's Responses to Questions

**Comments to the Author**

1. If the authors have adequately addressed your comments raised in a previous round of review and you feel that this manuscript is now acceptable for publication, you may indicate that here to bypass the “Comments to the Author” section, enter your conflict of interest statement in the “Confidential to Editor” section, and submit your "Accept" recommendation.

Reviewer #3: All comments have been addressed

2. Is the manuscript technically sound, and do the data support the conclusions?

Reviewer #3: Yes

3. Has the statistical analysis been performed appropriately and rigorously? 

Reviewer #3: Yes

4. Have the authors made all data underlying the findings in their manuscript fully available?

Reviewer #3: (No Response)

5. Is the manuscript presented in an intelligible fashion and written in standard English?

Reviewer #3: Yes

6. Review Comments to the Author

Reviewer #3: My prior concerns have been adequately addressed by the authors. Accordingly, the paper has improved, and I consider this version to be well developed.

As I noted in my previous review, this topic is interesting, and insights developed in this paper contribute to important deliberations in the literature.

7. PLOS authors have the option to publish the peer review history of their article (what does this mean?). If published, this will include your full peer review and any attached files.

Reviewer #3: No

---

## [Editor Report · Acceptance letter]

18 Mar 2024

PONE-D-23-19053R2 

PLOS ONE

Dear Dr. Wider, 

I'm pleased to inform you that your manuscript has been deemed suitable for publication in PLOS ONE. Congratulations! Your manuscript is now being handed over to our production team.

Kind regards, 

on behalf of

Dr. Ali B. Mahmoud 

Academic Editor

PLOS ONE